# Antennal Morphology and Fine Structure of Flagellar Sensilla in Hippoboscid Flies with Special Reference to *Lipoptena fortisetosa* (Diptera: Hippoboscidae)

**DOI:** 10.3390/insects13030236

**Published:** 2022-02-27

**Authors:** Annalisa Andreani, Antonio Belcari, Patrizia Sacchetti, Roberto Romani

**Affiliations:** 1Department of Agriculture, Food, Environment and Forestry (DAGRI), University of Florence, Piazzale delle Cascine 18, 50144 Firenze, Italy; annalisa.andreani@unifi.it (A.A.); antonio.belcari@unifi.it (A.B.); 2Department of Agricultural, Food and Environmental Sciences, University of Perugia, Borgo XX Giugno 74, 06121 Perugia, Italy; roberto.romani@unipg.it

**Keywords:** SEM, TEM, coeloconic sensilla, basiconic sensilla, arista, host location, *Lipoptena cervi*, *Hippobosca equina*, *Pseudolynchia canariensis*

## Abstract

**Simple Summary:**

In insects, host searching usually involves different kinds of stimuli, both visual and chemical, that may act in combination. External cues are perceived through specific sensory organs (sensilla), mainly present on the antennae. Understanding how ectoparasites belonging to the Hippoboscidae locate their hosts is crucial, since these flies infest animals and can attack humans, with veterinary and medical implications. The aim of this research was to study the antennae of four hippoboscid species, *Lipoptena cervi* (Linnaeus, 1758), *Lipoptena fortisetosa* Maa, 1965, *Hippobosca equina* Linnaeus, 1758, and *Pseudolynchia canariensis* (Macquart, 1840), investigating the morphology and the sensory structures present on these appendages. A typical conformation of the antennae with the envelopment of the third segment (flagellum) inside the first two have been observed. Moreover, two types of sensilla have been detected and their role in the perception of host odours and CO_2_ have been hypothesized. Other antennal structures seem to be involved in the detection of temperature and humidity variations. Our findings confirm that these hippoboscids use chemoreception for host location, giving insights into this complex process in this poorly investigated group.

**Abstract:**

*Lipoptena cervi* (Linnaeus), *Lipoptena fortisetosa* Maa, *Hippobosca equina* Linnaeus, and *Pseudolynchia canariensis* (Macquart) are hematophagous ectoparasites that infest different animal species and occasionally bite humans. Hosts are located by a complex process involving different kinds of stimuli perceived mainly by specific sensory structures on the antennae, which are the essential olfactory organs. General antennal morphology, together with distribution and ultrastructure of sensilla, have been studied in detail with scanning and transmission electron microscopy approaches. Observations have revealed some common features among the four studied hippoboscids: (a) typical concealment of the flagellum inside the other two segments; (b) characteristic trabecular surface of the flagellum; (c) peculiar external microtrichia; (d) presence on the flagellum of basiconic sensilla and grooved peg coeloconic sensilla; (e) unarticulated arista. The ultrastructure of *L. fortisetosa* revealed that microtrichia and the flagellar reticulated cuticle are not innervated. Different roles have been hypothesized for the described antennal structures. Microtrichia and the reticulated cuticle could convey volatile compounds towards the flagellar sensory area. Peculiar sensory neurons characterize the unarticulated arista which could be able to detect temperature variations. Coeloconic sensilla could be involved in thermoreception, hygroreception, and carbon dioxide reception at long distances, while the poorly porous basiconic sensilla could play a role in the host odour perception at medium–short distances.

## 1. Introduction

Hippoboscids are obligate hematophagous ectoparasites of vertebrates. These flies belong to the superfamily Hippoboscoidea together with other important families, such as Glossinidae (tsetse flies), Nycteribiidae, and Streblidae (bat flies) [1]. The family Hippoboscidae encompasses three subfamilies—Ornithomyinae, Hippoboscinae, and Lipopteninae—whose members live on birds, various mammal species, and ungulates, respectively [2,3]. Several representatives of these three subfamilies are well known for their veterinary and medical importance, since they can be responsible for diseases harmful to humans and animals [4].

Among members of the subfamily Ornithomyinae, *Pseudolynchia canariensis* Macquart (the pigeon fly) is a medium-sized species living especially on Columbiformes. It may transmit to its hosts the avian malaria parasite, *Haemoproteus columbae* Kruse, and, additionally, it can produce skin dermatitis in case of severe infestations [5]. *Hippobosca equina* L. (the forest fly), belongs to the Hippoboscinae and is an ectoparasite mainly of horses and donkeys, on which it can cause several annoyances and skin injuries. This species can also act as a vector of pathogens dangerous both to animals and humans, such as *Anaplasma* spp. [6], *Corynebacterium pseudotuberculosis equi* [7], and *Bartonella chomelii* Maillard et al. 2004 [8]. Within the subfamily Lipopteninae, *Lipoptena cervi* L. and *L. fortisetosa* Maa (the deer keds) predominantly attack cervids, on which they can cause skin diseases and behaviour alterations in cases of high parasite population density [9,10]. Moreover, they can play an important role as carriers of pathogens, mainly *Anaplasma* spp., *Bartonella* spp., *Borrelia* spp., *Coxiella* spp., *Theileria* spp., and *Trypanosoma* spp. [11,12,13,14,15,16,17]. Recently, the Asian species *L. fortisetosa* has colonized most European countries, including Italy, where it is competing with *L. cervi* for territories and host microniches [18].

The four above-mentioned hippoboscid flies live at the expense of a few host species, wherewith they established a strict interaction depending on the host behaviour and morphology. Due to this specialized parasitic life, these flies display extreme specialization of many features, such as a flattened body, robust legs equipped with claws which allows it to tightly cling to the host’s coat, and a prognathous head, which allows it to firmly adhere to the host [19]. Antennae show a remarkable morphological adaptation with the scape and pedicel fused together in almost all species [20]; moreover, the flagellum is housed within a cavity formed by the first two antennal segments [19,21]. Antennae are almost completely hidden inside two deep hollows, named antennal sockets or fossae. These sensory appendages play a primary role in the host location in hematophagous dipterans, and are responsible for the detection of odour cues. This behavioural aspect has been demonstrated in members of the suborder Nematocera, such as black flies and mosquitoes [22,23], as well as in representatives belonging to the suborder Brachycera, such as muscids, tabanids, and glossinids [22]. Antennal sensory structures of hippoboscid flies have been studied in *Hippobosca equina*, *H. longipennis* Fabricius, and *Melophagous ovinus* L., where the external surface of the flagella have different kinds of sensilla [21]. Given the concealment of the flagellum and the reduction in the other antennal segments, Zhang et al. [21] speculated that these modifications may have caused the lack of the primary sensory function (olfaction), defining these flies as “inactive ectoparasites”. Actually, soon after their emergence from puparia, most of hippoboscid species spend the pre-parasitisation period resting on vegetation or flying, using different kinds of stimuli to locate a suitable host. In Finland, *L. cervi* adults are able to survive without feeding for over a month [24], so that the host-seeking period may be extended. These insects require remarkable energy expenses to detect external signals, mainly odour cues. Host location in hippoboscid flies needs to be further investigated, especially in those species in which newly emerged adults occur in areas with no availability of hosts, leading to an active host searching mediated by external stimuli, which are especially detected by antennal sensilla. In *P. canariensis*, *H. equina*, *L. cervi*, and *L. fortisetosa*, the antennal sensory structures present on the external segments display peculiar morphological adaptations, since these species have diverse parasitic behaviours and a different association level with their hosts [19].

The present paper deals with a morphological analysis of antennal structures and sensory patterns in these four species, with special reference to the deer ked *L. fortisetosa*, which has been investigated for the sensillar ultrastructure. These observations may contribute to a better understanding of the host location process of these hematophagous ectoparasites.

## 2. Materials and Methods

### 2.1. Insect Collection

Hippoboscid flies were collected in several areas of Tuscany (central Italy) for Scanning Electron Microscope (SEM) observations. Wingless deer keds were manually picked up by cervid skin pieces provided by hunters during the culling season of 2019–2020. Specimens of *P. canariensis* were collected from pigeons during an official wildlife surveillance program performed by Provincial Wildlife Police in San Miniato (Pisa), while *H. equina* adults were picked up from horses in a stable in Marradi (Firenze). For Transmission Electron Microscopy (TEM), winged adults of *L. fortisetosa* were collected in a wooded area in Schignano (Prato) at about 550 m a.s.l. (43.967432 N; 11.101761 E). Winged adults were caught by sweeping, maintained in microvials containing a small piece of cotton soaked with water and sugar, and kept at low temperature (4–6 °C) for a few days, pending TEM analysis.

### 2.2. SEM Procedures

All hippoboscid adults (at least 20 specimens each species, about 60 for *L. fortisetosa*) were anaesthetized at −20 °C for 20 min and then maintained in 70% ethanol pending preparation procedures. Specimens were removed from ethanol, rinsed with distilled water several times, and then sonicated for 15 min in a 10% potassium hydroxide (KOH) distilled water solution to remove impurities and secretions from their bodies. After that, the samples were rinsed again in distilled water to remove KOH residues. Subsequently, adults were dehydrated in a series of graded ethanol concentrations (from 70% to 90% with 10% increasing in each concentration, then 95% and 99%, for 10 min in each concentration). Antennae were excised from the heads and dissected to extract the internal flagella. Then, all samples were air-dried, mounted on aluminium stubs and gold-coated with a sputter coater device (S150B; BOC Edwards, Burgess Hill, UK). SEM observations were made using an FEI Quanta 200 high vacuum, low vacuum, and environmental scanning electron microscope (Thermo Fisher Scientific, Inc., Waltham, MA, USA) at the Department of Agriculture, Food, and Environment (DAFE), University of Pisa, and a Zeiss Evo 40 at the centre “Centro di Servizi di Microscopia Elettronica e Microanalisi” (MEMA), University of Florence. The morphology and the external sensillar pattern of the antennae were examined and described according to the terminology and nomenclatures reported by Maa and Peterson [20] and Zhang et al. [21].

### 2.3. TEM Procedures

Ten live winged adults of *L. fortisetosa* were CO_2_ anaesthetised and thereafter immersed in a glutaraldehyde/paraformaldehyde solution (2.5% in 0.1 M cacodylate buffer +5% sucrose, pH 7.2–7.3) for 3–4 h. The antennae of some specimens were isolated from the rest of the head capsule to reduce the size of the tissue to be fixed and to facilitate fixative penetration. However, full heads were also processed. After the first fixation step, samples were rinsed twice in 0.1 M cacodylate buffer (15 min each step) and kept at 4 °C overnight. Then, samples were post-fixed in a 1% osmium tetroxide solution (OsO_4_) for about 50 min. After rinsing with the same buffer, specimens were then dehydrated in a graded series of ethanol (from 50 to 90% with 10% increasing concentration each, then 95% and 99%), with each step lasting 15 min. Subsequently, specimens were exposed to pure propylene-oxide, then to a 50/50 blend of propylene oxide and Epon–Araldite resin to improve resin infiltration. Each sample was finally infiltrated with an Epon–Araldite resin and incubated at 65 °C for 48 h. Embedded samples were sectioned using a diamond knife (Drukker) using a Bromma ultramicrotome (LKB, Stockholm, Sweden). Ultrathin sections (60–90 nm) were collected using formvar coated, 50 mesh copper grids, and then stained with uranyl acetate (20 min at room temperature) and lead citrate (5 min at room temperature). Grids were investigated with a Philips EM 208 TEM (Thermo Fischer Scientific, Hillsboro, OR, USA). Digital photographs (1376 × 1032 pixels, 8 bit, uncompressed greyscale TIFF files) were obtained using a high-resolution digital camera MegaView III (SIS, Muenster, Germany) connected to the TEM. TEM data were obtained at the “Centro Universitario di Microscopia Elettronica e Fluorescenza (CUMEF; Università degli Studi di Perugia, Italy)”.

## 3. Results

The four studied species show a similar arrangement of the antennal pattern with antennae inserted inside peculiar head cavities, named fossae or antennal sockets. Except for *P. canariensis*, the scape and pedicel are fused and house the third segment, the flagellum.

### 3.1. Pseudolynchia canariensis

The outer part of *P. canariensis* antennae, depicted in Figure 1A–C, displays the arrangement of these appendages protruding externally from the socket with the scape articulated with both the lunula and the fronto-clypeus and the proximal part of the pedicel, to some extent, fused with the scape (Figure 1A,B).

Several long bristles constitute the external sensillar apparatus of the first two segments, which have been described in a previous paper [19]. Moreover, the pedicel surface adjacent to the fronto-clypeus is partially covered by microtrichia. An unbranched arista, with a shovel-shaped tip (Figure 1C,D), originates from the dorsolateral part of the introflexed flagellum, which is pear-shaped (Figure 2A). The flagellum is marked by an irregular surface of dense and long microtrichia mixed with cuticular trabeculae, uniformly arranged and forming several pits located on the dorsolateral area (Figure 2). Within these hollows, different kinds of receptors, mainly coeloconic grooved and a few basiconic sensilla, are interspersed.

### 3.2. Hippobosca equina

In this species, the antenna lies in the antennal socket with only the dorsal surface externally exposed and entirely covered by microtrichia except for a small area on the top (Figure 3A,B). Three mechanosensory bristles are present on the distal part of the segment [19]. Additionally, a small furcate arista protrudes in the ventral region of the hollow which is wallpapered by a dense covering of microtrichia (Figure 3C,D). Figure 4A shows the piriform flagellum with a non-articulated arista on the dorsoanterior area. The surface of the flagellum is covered by a reticulated cuticle from which short microtrichia rise up. These microtrichia are shorter and display a wider base than those of *P. canariensis* (Figure 4B). However, similarly to the cuticular pattern of the pigeon fly, several sensilla are located inside sensory pits on the dorsolateral region. Coeloconic grooved sensilla, almost always sunken in sensory pits, are mainly spread in the proximal part of the flagellum, while multiporous basiconic sensilla occur around the arista (Figure 4A,C,D).

### 3.3. Lipoptena cervi

The external edge of *L. cervi* antennae presents different types of sensilla previously described [19]. The external surface shows a typical microtrichia overlay which thickens approaching the pedicel opening (Figure 5), where, interestingly, microtrichia become furcate with two or three prongs (Figure 5D). A non-articulated branched arista protrudes from the pedicel hollow (Figure 5A,C,D). The piriform flagellum displays the typical trabecular surface in the proximal part (Figure 6A). Close to the arista, the cuticle forms shallow depressions in which microspines are present. Similar to the previously described species, *L. cervi* shows several sensory pits on the proximal part of the flagellum, characterized by the presence of basiconic and coeloconic grooved sensilla (Figure 6B–D).

### 3.4. Lipoptena fortisetosa

In this fly, the antennal apparatus differs externally from those of the other described species by the presence of a series of peculiar aligned robust mechanosensory bristles on the edge of the pedicel (Figure 7A) [19]. The pedicel is bean-like and bears sparse microtrichia on the dorsolateral part, toward the fronto-clypeal area; the fan-shaped tip of the arista emerges from the hollow (Figure 7B and Figure 8A). The antenna is housed inside a deep antennal fossa which encloses almost all the segment surface (Figure 7C). As in *L. cervi*, the proximal region of the flagellum is covered by a trabeculated surface with sensory cavities in the dorsal area (Figure 7D). This trabecular organization of the surface appears even more evident in serial cross-sections of the antenna performed at the very distal region of the flagellum, as well as more proximally (Figure 8B,C). TEM cross-sections show shallow cavities occupied by cuticular pegs and sparse basiconic and coeloconic sensilla, while clustered basiconic sensilla are housed in deeper invaginations of the flagellum cuticular wall (Figure 8D,E). TEM investigation reveals the following internal organization for the above reported sensory structures.

The long bristles located at the external margin of the pedicel are characterised by long cuticular shafts, straight and sharply pointed (Figure 7A and Figure 9A). The shaft base is housed inside a rounded socket and exhibits sturdy external grooves running from the base to the tip. Serial cross and longitudinal sections reveal the presence of a single sensory neuron with a large cell body containing a prominent nucleus and a relatively short inner dendritic segment (Figure 9D). Right below the socket, from the inner dendritic segment an outer dendritic segment is differentiated (Figure 9C), with a typical ciliary constriction region. The outer dendritic segment ends at the base of the shaft with an electron dense tubular body enveloped by the dendritic sheath (Figure 9B). The cuticular shaft is made of thick cuticle with a small lumen located in the centre, where no dendrites or dendritic branches are found. The thinner and shorter microtrichia present on the dorsolateral side of the pedicel are not innervated (Figure 9E).

The arista is short, slightly curved, and can be divided in two distinct parts of the same length: a proximal portion, stick-like, that is connected to the flagellum; and a distal region that is enlarged and fan-like (Figure 10A). The flattened distal region examined through TEM shows no lumen (Figure 10B,C). The cross-section of the proximal part of the arista displays a lumen filled with extracellular material (Figure 10D). It is noteworthy that externally the arista does not show any specialized cuticular structures that could be related to a sensory function. Cross-sections of the arista’s basal region, connected with the pedicel, reveal the absence of a specialised socket, while the arista lumen contains a cluster of cells located eccentrically, close to the wall. In this area (Figure 10E,F), two groups of sensory neurons have been highlighted: the first one formed by two outer dendritic segments and the second one comprising three outer sensory neurons. In both cases, the grouped sensory neurons are embedded by dendrite sheaths.

Basiconic sensilla (BS) can be mainly found inside the deeper cavities that are spread on the dorsal area of the flagellum, as well as on the surface of the flagellum itself. BS are characterized by the presence of an external cuticular peg standing on the antennal wall and surrounded by the trabeculated cuticle that covers most of the flagellum surface (Figure 11A). The representative basiconic sensillum shows a blunt tip and pores on the sensillum wall. Longitudinal sectioning reveals a cuticular shaft made of a thin cuticle perforated by numerous pores distributed on the distal half (Figure 11B–D). The shaft is inserted on the antennal wall without a flexible socket (Figure 11B). Internally, a single sensory neuron innervates each sensillum. The outer dendritic segment enters the proximal part of the sensillum, from where several dendrite branches arise, filling the shaft lumen for all its length (Figure 11B–D). Cross-sections under the sensillum base show the sensory neuron enclosed by a well-defined dendrite sheath and the thecogen cell (Figure 11E–G). The dendrite sheath embeds the outer dendritic segment of the sensory neuron also inside the cuticular shaft, up to the level where the neuron starts to branch (Figure 11B).

Coeloconic sensilla (CS) are grooved pegs interspersed on the flagellum surface, often accompanied by basiconic sensilla. Also in this case, the presence of coeloconic sensilla seems to be restricted to the dorsal surface of the flagellum. These sensilla have a typical organization with a short cuticular shaft positioned inside a shallow depression. The shaft displays several grooves on its distal half, due to cuticular ridges that give to the sensillum a unique morphology (Figure 12A,B). TEM cross-sections reveal a distal star-shaped structure (as a result of the external ridges) with nine–ten spikes (Figure 12C). In between the ridges, there are spoke channels that connect the internal lumen with the external environment. The peg lumen shows two–three dendrites (Figure 12C). Proximally, the cuticular shaft exhibits the typical double-walled organization delimiting an innermost lumen, housing two–three dendrites, and an outermost space (Figure 12D). Under the socket, a thick dendrite sheath embeds the associated outer dendritic segments (two–three per sensillum), with a bulk of electron lucid vesicles present in this region (Figure 12E). Sections carried out more proximally show dendrites still embedded by the dendrite sheath, but this appears less thick and tight (Figure 12F).

## 4. Discussion

The antennal apparatus of the described hippoboscids reveals a general similar arrangement in the structure of the segments in both sexes. In fact, because of their highly specialized parasitic lifestyle the main sensory area, the flagellum, is concealed inside the pedicel in order to protect the sensory area during the movements of the ectoparasites within the host coat. Generally, among hippoboscids the scape is not well recognizable, being partially or completely fused with the lunula [20]. Otherwise, it is also possible that this segment is not completely absent, but fused with the pedicel, as postulated for species of Nycteribiidae [25]. This hypothesis may be acceptable since in several other hippoboscid species, especially belonging to the subfamily Ornithomyinae, the three antennal segments are clearly separate. For example, in *P. canariensis,* the scape is distinguishable and partially articulated with the fronto-clypeal region and the lunula, as also seen in the Nearctic louse flies *Olfersia fumipennis* (Sahlberg, 1886), *Ornithoica vicina* (Walker, 1849), *Ornithoctona erythrocephala* Leach, 1817, and *Icosta americana* (Leach, 1817) [20]. The articulation between the first two segments is likely due to the different host selection pressure that louse flies received during their adaptive process in comparison with other hippoboscid species living on mammals. In fact, the partial or total fusion between the scape and the pedicel, together with the introflexion of the flagellum, likely allowed the flies to reduce the antennal surface and consequently to mechanically protect these appendages inside the antennal sockets or fossae. As already mentioned, this morpho-functional organization protects the antennal sensory function of hippoboscids in a dense and harsh environment, such as the animal coat [19].

The morphology and location of antennae in Nycteribiidae and Streblidae (Hippoboscoidea) are quite similar to those of the four species studied in this paper. Representatives of these two families live on bats, and, as do other hippoboscid species, establish a strong parasitic association with their hosts [26,27]. In fact, Nycteribiidae and Streblidae species display the reduction or absence of the first antennal segment, the scape, and the complete or partial introflexion of the flagellum inside the previous segment. However, in nycteribiids, the piriform flagellum differs from that of hippoboscids in the surface pattern and for the sensory pits, only in terms of position and number. In fact, the flagellar surface appears reticulose and bears only four sensory pits located on the dorsodistal part [28]. In contrast, in Streblidae the flagellum is different from nycteribiids, since it is not completely encapsulated inside the pedicel and shows a bilobate shape with a surface covered by minute spines [29], which are similar in arrangement to those observed in the described hippoboscid species.

Into the superfamily Hippoboscoidea, representatives of Glossinidae (tsetse flies) show a very different antennal apparatus compared with Hippoboscidae, Nycteribiidae, and Streblidae. In fact, Glossinidae are free-living parasites which do not establish a permanent association with the host, with antennae more similar to Muscomorpha [30]. In these hematophagous flies, antennae are always three-segmented with a well-developed external flagellum bearing a long arista which originates from the dorsoproximal part of the segment [31]. Thus, the arrangement of the antennae in tsetse flies is similar to those of higher dipterans, but these appendages are housed in a deeper antennal socket [32], which presumably protect the flagella during the trophic activity. This arrangement is comparable to that of Hippoboscoidea and other parasites of veterinary importance, e.g., in the muscid flies *Haematobia irritans* (Linnaeus, 1758) [33] and *Hydrotaea irritans* (Fallén, 1823) [34]. Additionally, in hippoboscid species, the lack of the external flagellum may have led during the evolution process to the development of these sockets which could act as a funnel, directing external volatile compounds towards the sensilla located on the introflexed flagellum.

Noteworthy are the microtrichia which densely cover the antennal hollow of the four hippoboscid flies. These processes have been observed for their ultrastructure on *L. fortisetosa* and are not innervated. For this reason, microtrichia do not play a primary role in the sensory perception but, being furcated in two or three branches and forming a kind of dense carpet of hairs, could be involved in directing odours, conveying them towards the internal sensory area on the flagellum. In particular, in *H. equina* the microtrichia are present also on the internal surface of the antennal fossa, lending support to this hypothesized role. Recently, six types of microtrichia, including branched ones, have been detected also on the flagellum of the tabanid fly *Haematopota pandazisi* (Krober, 1936) by Pezzi et al. [35], who postulated the role of these structures, together with different kinds of sensilla, in the sensory perception.

The introflexed flagellum is the main sensory area of the antenna, and its surface is covered by trabecular structures which resemble the above discussed microtrichia. Similarly, this reticulate surface could serve to maintain the odour cues, allowing porous chemosensilla, which are mainly present inside the sensory pits, to improve the perception of volatile stimuli. The assortment of flagellar sensilla is very similar in the four studied species since they display only two types of sensory structures: grooved coeloconic and basiconic sensilla. Their arrangement on the flagellum is diverse, since grooved sensilla are more abundant and more present in the pits on the dorsodistal part of the flagellum, while basiconic sensilla are fewer and are mainly distributed in the anterior part of the segment, around the base of the arista. Additionally, the number and the arrangement of these two types of sensilla is different among the investigated flies. *Hippobosca equina* seems to display a higher number of sensory pits compared with the other three species, with most of these depressions housing coeloconic sensilla. In general, the richness of sensilla is related to the different lifestyle of ectoparasites; in fact, a permanent ectoparasite living in close association with its host does not require many receptors, while a temporary ectoparasite needs a major number of sensilla to frequently locate a new host [36,37]. For instance, *H. equina* has a higher number of sensilla compared with the other three hippoboscid species, and is not strictly associated with a single host specimen. In *Glossina morsitans*, flagellar sensilla show a more complex pattern compared with that of hippoboscid flies, displaying four types of sensilla: basiconic, coeloconic, trichoid, and intermediate sensilla [38]. Trichoid sensilla seem to be involved in sex pheromone transduction as demonstrated in *Drosophila melanogaster* [39,40]. The absence of this type of sensillum in the investigated hippoboscid flies, may be explained by the mating behaviour of these dipterans. They mate usually when they have colonized the host microniches, although some species can mate on the wing [37]. In fact, since the wingless adults of both sexes live aggregated on the host body, it is possible to hypothesize that they do not require any pheromone to locate a partner.

In these ectoparasites, the prevalence of coeloconic sensilla compared with basiconic sensilla may be attributed to their function as chemoreceptors. In *D. melanogaster*, coeloconic grooved sensilla are involved in the detection of ammonia, ketones, and amines [41]; moreover, these kinds of sensilla are able to perceive humidity variations, but not temperature changes [41]. Similarly, in *Anopheles gambiae* these sensilla are involved in ammonia detection [42]. However, coeloconic sensilla have also been classified as thermoreceptors, hygroreceptors, and carbon dioxide receptors in other orders of insects [43]. Further, a thermoreception role has been hypothesized in *Aedes aegypti* L. [44], and an olfactory and hygroreceptive function have been proposed for *Culicoides furens* (Poey) [45], which are hematophagous dipterans of medical and veterinary importance. These supposed functions may also be similar for hippoboscid flies. As a matter of fact, a previous work conducted in field using people with two heated bags to one shoulder and two cold bags to the other shoulder, showed that *L. cervi* winged adults usually landed on the hotter part, demonstrating that they were attracted to and perceived temperature at short distance [46]. As well, in a lab experiment carried out in arenas, two species of Nycteribiidae, *Penicillidia conspicua* Speiser, 1901 and *P. dufourii* (Westwood, 1835), were found to be more active moving more often towards the heat source; although they responded more strongly to the combination of carbon dioxide and heat stimuli [47].

Regarding basiconic sensilla present on the flagellum, our investigations showed that their number is lower compared with that of coeloconic sensilla. This is consistent in all the observed species, and mainly in *H. equina*, where the distribution of these sensilla has been mapped, highlighting the higher abundance of coeloconic compared with basiconic sensilla [21]. Basiconic sensilla occur in higher density on the flagellum of Glossinidae flies [38], as well as of other hematophagous dipterans, such as *Stomoxys calcitrans* [48]. Similarly, basiconic sensilla are more represented in some saprophagous species [49,50,51,52], as well as in phytophagous dipterans like fruit flies [53,54]. Although there are a few electrophysiological studies about the role played by basiconic sensilla, it is known that they are mainly involved in odour detection due to the presence of many pores on the external wall [55]. The multiporous basiconic sensilla occurring on *L. fortisetosa* differ from those described in other dipterans [34,35,54,56], since the ultrastructure shows a reduction in the presence of the wall pores, which occur only in the distal half of the shaft. The limited number of pores on the basiconic walls could be due to the perception of the host odours which is activated just when the parasite is approaching the host at short–medium distances. On the other hand, previous studies conducted on different families of hematophagous dipterans revealed that the main stimuli activating the response at long distances are carbon dioxide, ammonia, or other volatile substances [22,44,45]. In fact, electrophysiological studies demonstrated that neurons associated with coeloconic sensilla were activated by different kinds of external stimuli, such as CO_2_, temperature, and humidity, in many species of insects [57,58]. A relevant feature of the basiconic sensilla in *L. fortisetosa* is the presence of a single sensory neuron that innervates each sensillum. The occurrence of a relatively low number of neurons associated with multiporous sensilla is reported for other species belonging to different insect orders. In the planthopper *Hyalestes obsoletus* Signoret, a grooved peg sensillum coeloconicum has been observed at the level of the expanded base of the thread-like flagellum. This sensillum is innervated by a single sensory neuron that is highly branched inside the sensory peg, for which a role in CO_2_ perception is hypothesized [59]. Within Diptera, sensilla basiconica are described in detail in *D. melanogaster*, where different types of sensilla are present: small sensilla basiconica (innervated by two sensory neurons) and large sensilla basiconica (innervated by up to four sensory neurons) [60,61]. In *D. melanogaster*, such diversity is related to the high antennal sensitivity to volatile compounds and the gradient distribution pattern of antennal sensilla on the funiculus. In *L. fortisetosa*, we hypothesize that the great reduction in number and size of sensilla basiconica, as well as in the number of associate sensory neurons, could be linked to the reduced range of volatiles exploited during intra- and interspecific interactions.

The arista was unarticulated in all the four investigated hippoboscids. This structure shows a remarkable difference in the morphology of the distal part, being furcate (*L. cervi*), fan-shaped (*L. fortisetosa*), branched (*H. equina*), or spatulate (*P. canariensis*). According to our knowledge, currently, the fusion of the arista with the flagellum is highlighted for the first time in dipterans. This arrangement may be due to the particular adaptation evolved during the introflexion of the flagellum inside the pedicel. Although the arista tips are so diverse, our investigations performed on *L. fortisetosa* revealed that no sensilla on the external surface, nor are cuticular pores present. The lack of external sensory structures differs from that observed on the arista of the human bot fly, *Dermatobia hominis* (Linnaeus Jr. in Pallas, 1781), which showed different kinds of receptors, such as long bristles, coeloconic, and styloconic sensilla [62]. Additionally, on the arista (properly termed as stylus) of the marsh fly *Sepedon fuscipennis* Loew 1859, mechanosensilla arranged differently in females and males may have possible functions related to mating and foraging behaviour [63].

Ultrastructural investigations on *L. fortisetosa* showed that two bundles of sensory neurons are present close to the base of the arista, with an apparent lack of connection with the external cuticle. Similar findings have been reported for *D. melanogaster*, *Calliphora erythrocephala* (Masquart, 1834), and *Musca domestica* Linnaeus, 1758. Here, the arista ultrastructure is much more complex compared with *L. fortisetosa*, and houses a variable number of aberrant sensilla that lie freely in the haemolymph, with a possible thermoreceptive function [64]. It is noteworthy that the number of these unusual sensilla increases with the increase in the arista size, and this could explain the low number of sensory neurons recorded in *L. fortisetosa* (five) compared with the amount found in *D. melanogaster* (six), *M. domestica*, and *C. erythrocephala*, (24 and 36, respectively) [64]. It is conceivable that the aristal sensilla in *L. fortisetosa* could be involved in the perception of temperature variation, a key factor in the detection and discrimination of the warm-blood hosts exploited by these ectoparasites.

Sensillar pattern of the studied hippoboscid flies reveals how the host location process in these ectoparasites is quite complex; in fact, the abundance of coeloconic grooved sensilla supports the long-distance activation of the newly emerged winged adults, which should be mainly stimulated by ammoniacal substances and carbon dioxide. Additionally, we can speculate that, once the principal odour stimulus is detected, other factors may guide the adult towards the host. Heat and colour take part in the host location at medium and short distances, as reported for *L. cervi* [46] and *L. fortisetosa* [65].

## 5. Conclusions

Morphological investigations carried out by SEM and TEM on four hippoboscid species revealed a strong adaptation in the antennal apparatus due to the parasitic lifestyle of these flies. In particular, the main sensory area, the flagellum, is concealed inside the pedicel. This latter is fused with the first antennal segment, the scape, in *L.*
*cervi*, *L. fortisetosa* and *H. equina*, while it is partially articulated in *P. canariensis*. The arista appears differently shaped in the species studied and is not articulated with the flagellum, which is a unique feature in dipterans. The flagellum bears two different types of sensory structures: grooved coeloconic and basiconic sensilla. The number and the arrangement of these sensilla is quite different among the species according to their life cycle and association level with the hosts. However, a prevalence of coeloconic sensilla has been highlighted in all the investigated hippoboscids. These structures are generally involved in volatile detection, such as CO_2_, ammonia, and other odours, but they can also play a role in perceiving changes in humidity and temperature. Similarly, the arista could be involved in the detection of temperature variations, since, in *L. fortisetosa*, it houses peculiar sensory neurons.

Finally, we hypothesize that locating hosts at medium and long distances in winged adult hippoboscids occurs mainly due to these sensilla; although, it is a complex process that involves visual stimuli as well. Since basiconic multiporous sensilla are present in few numbers and display a reduction in the abundance of wall pores along the shaft, they probably play a role in the host location at medium–short distances. However, further experiments, such as electrophysiological and behavioural bioassays, are needed to confirm these hypotheses.

## Figures and Tables

**Figure 1 insects-13-00236-f001:**
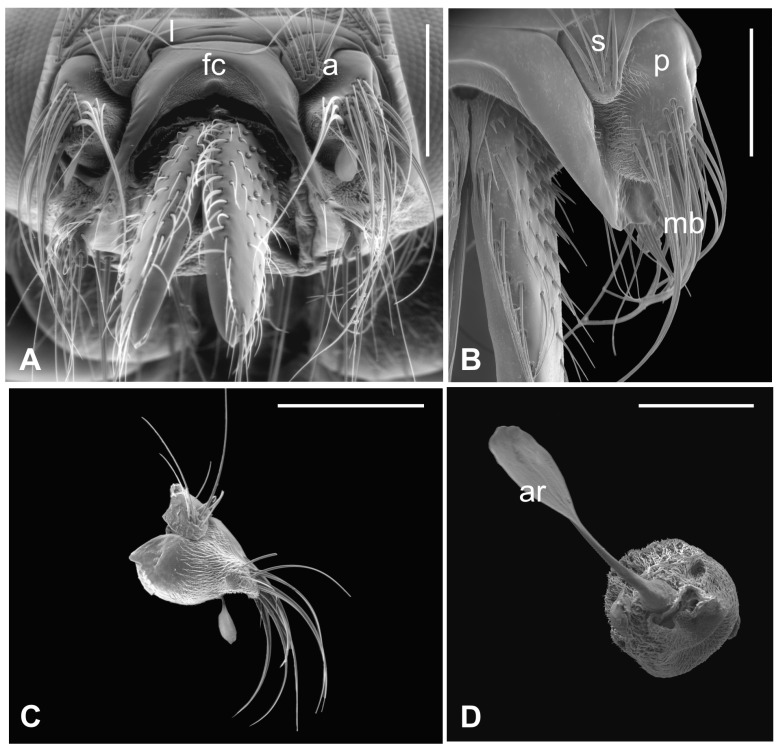
*Pseudolynchia canariensis*. (**A**) Frontal view of the head with antennae (a), which are partially fused with the lunula (l) and the fronto-clypeus (fc); (**B**) dorsal view of the antenna with the visible articulation between the scape (s) and the pedicel (p), bearing long bristles (mb) with probable mechanosensory function; (**C**) antenna excised from the antennal socket, showing mechanosensory bristles and the protruding arista; (**D**) flagellum with the non-articulated, shovel-shaped arista (ar). Bar scale: (**A**) 300 µm; (**B**) 200 µm; (**C**) 400 µm; (**D**) 100 µm.

**Figure 2 insects-13-00236-f002:**
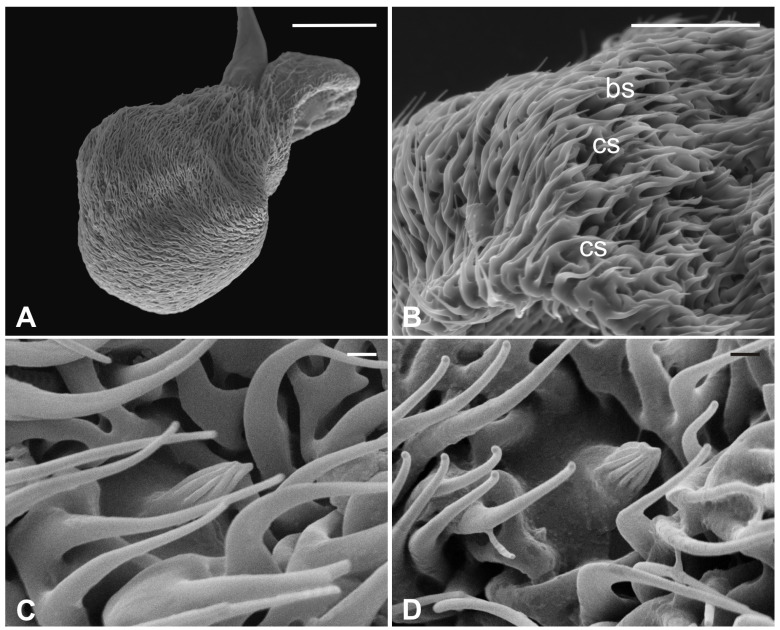
*Pseudolynchia canariensis*. (**A**) Lateral view of flagellum with cuticular pits housing mainly grooved coeloconic sensilla and rare basiconic sensilla. Note the characteristic surface arrangement; (**B**) detail of the typical trabecular structure of the flagellum, with coeloconic (cs) and basiconic (bs) sensilla accommodated in pits; (**C**–**D**) magnification of coeloconic grooved sensilla with different features. Bar scale: (**A**) 50 µm; (**B**) 20 µm; (**C**,**D**) 1 µm.

**Figure 3 insects-13-00236-f003:**
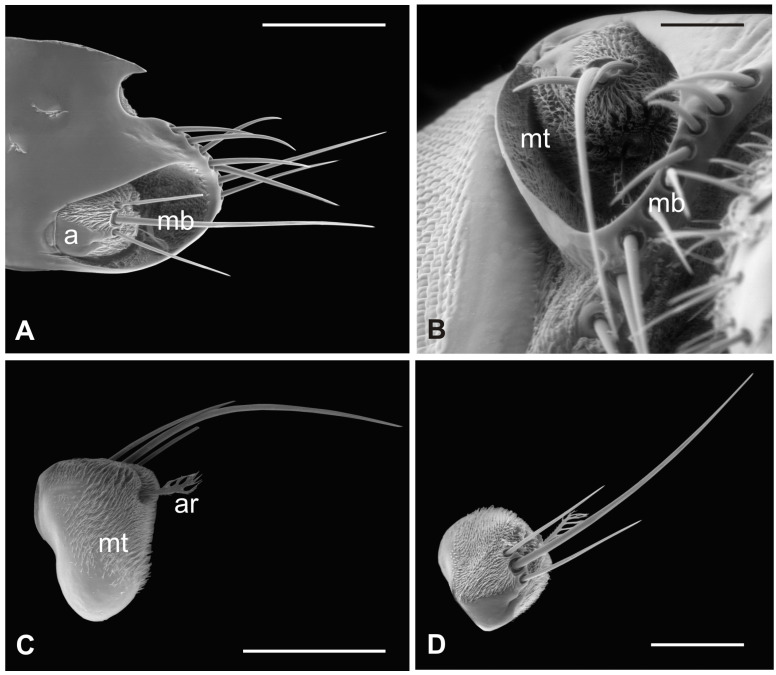
*Hippobosca equina*. (**A**) Dorsal view of antenna housed in the large antennal fossa, bearing three mechanosensory sensilla (mb). Note the long bristles (mb) probably with mechanosensory function at the edge of the structure; (**B**) ventral view of the antenna with the pedicel partially covered by a dense layer of microtrichia (mt) present also on the wall of the antennal fossa; (**C**,**D**) lateral and dorsal view of the antenna showing microtrichia (mt) and the typical branched arista (ar). Bar scale: (**A**) 300 µm; (**B**) 100 µm; (**C**) 300 µm; (**D**) 200 µm.

**Figure 4 insects-13-00236-f004:**
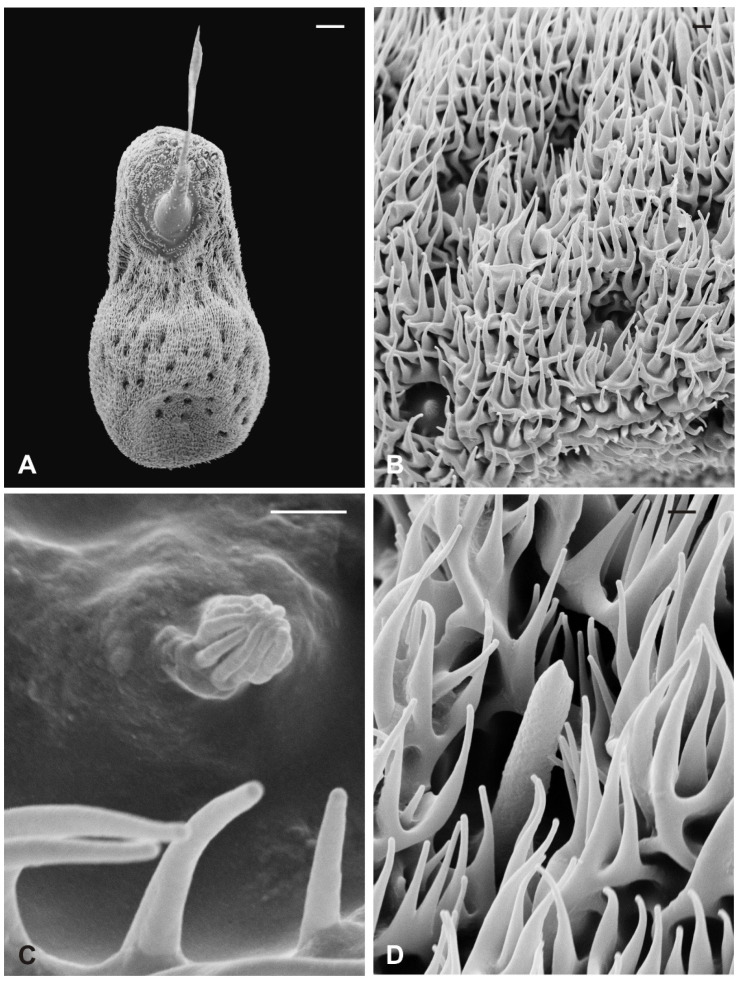
*Hippobosca equina*. (**A**) Dorsal view of the piriform flagellum with numerous sensillar pits located mostly on the dorsoproximal part. Note the non-articulated arista with a large base; (**B**) magnification of the surface characterized by differently sized microtrichia with coeloconic grooved sensilla accommodated inside cuticular depressions; (**C**) high magnification of a coeloconic grooved sensillum with evident finger-like projections; (**D**) high magnification of a basiconic sensillum embedded within microtrichia and cuticular trabeculae. Bar scale: (**A**) 20 µm; (**B**) 2 µm; (**C**,**D**), 1 µm.

**Figure 5 insects-13-00236-f005:**
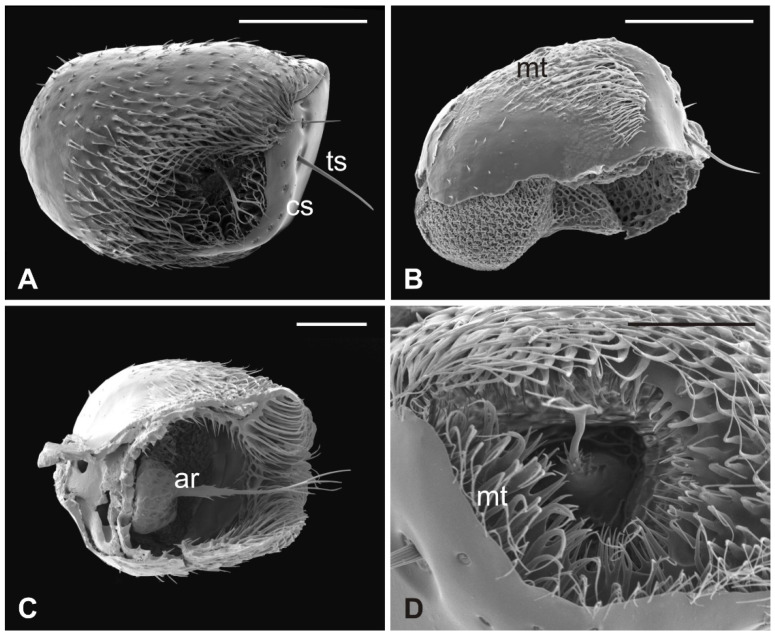
*Lipoptena cervi*. (**A**) Ventral view of the antenna with arista protruding from the hollow densely covered by microtrichia. Note the coeloconic (cs) and trichoid (ts) sensilla present on the pedicel edge; (**B**) lateral view of the opened pedicel with microtrichia (mt) showing the introflexed flagellum with the characteristic trabecular surface; (**C**) ventrolateral view of the dissected pedicel displaying the anterior part of the flagellum from which the branched arista (ar) originates; (**D**) antennal hollow magnification showing the arista with furcate microtrichia (mt). Bar scale: (**A**,**B**), 100 µm; (**C**) 50 µm; (**D**) 40 µm.

**Figure 6 insects-13-00236-f006:**
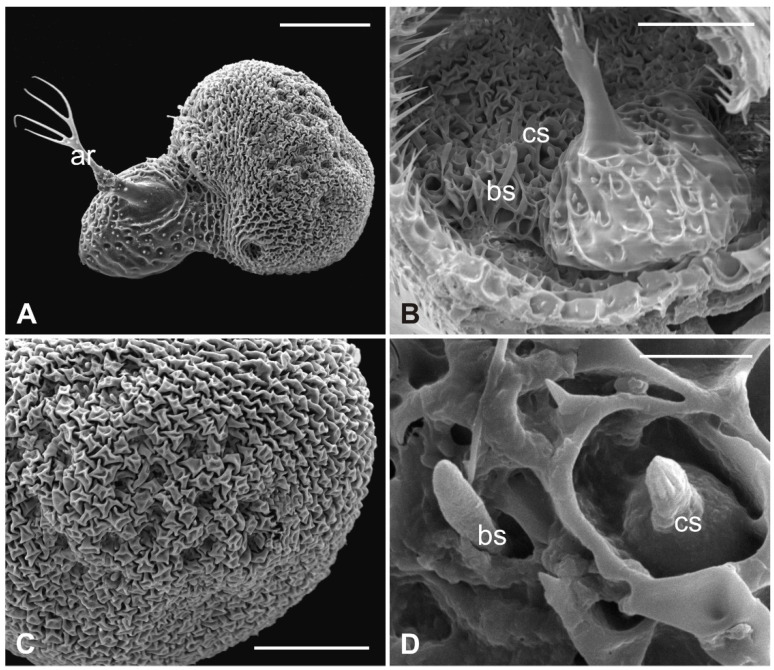
*Lipoptena cervi*. (**A**) Dorsolateral view of the flagellum showing a typical trabecular surface and the non-articulated arista (ar) with the branched tip. Note the cuticular depressions housing sensilla; (**B**) magnification of the sensory area close to the arista base with visible coeloconic (cs) and basiconic (bs) sensilla; (**C**) magnification of the dorsodistal part of the flagellum showing trabeculae and sensory pits; (**D**) magnification of a multiporous basiconic (bs) sensillum and a grooved coeloconic sensillum (cs). Bar scale: (**A**) 50 µm; (**B**,**C**) 30 µm; (**D**) 5 µm.

**Figure 7 insects-13-00236-f007:**
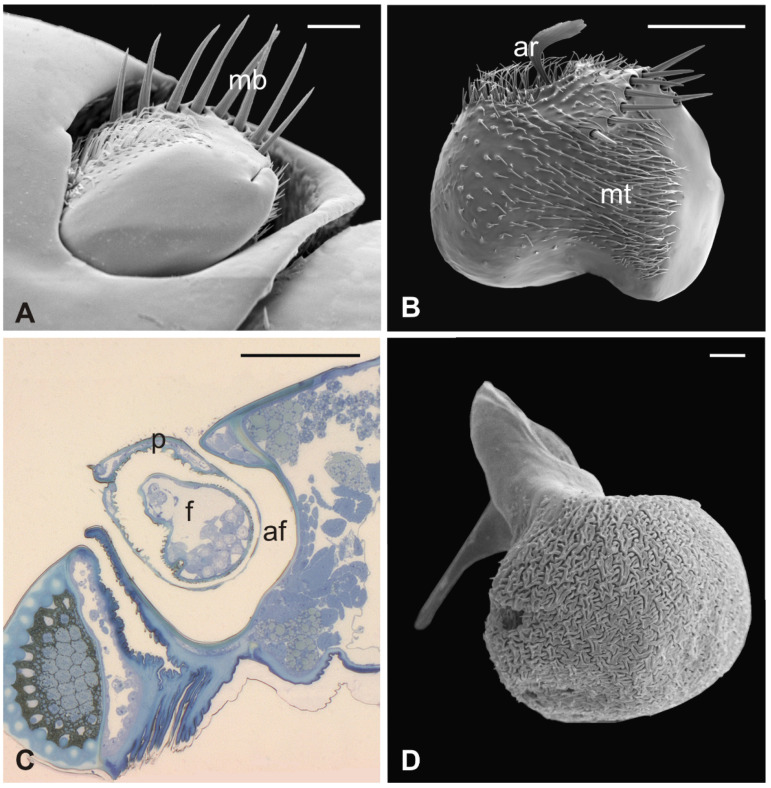
*Lipoptena fortisetosa*. (**A**) Dorsal view of the antenna with the typical aligned mechanosensory bristles; (**B**) lateral view of the bean-like pedicel showing sparse microtrichia (mt) thickening close to the hollow from which the arista (ar) protrudes with a spatulate tip; (**C**) light microscopy cross-section of the head showing the deep and large antennal fossa (af), where the pedicel is inserted. Inside the pedicel (p), it is possible to observe the pear-like flagellum (f) with clustered nuclei; (**D**) lateral view of the piriform flagellum with the typical reticulate surface covering the proximal part of the segment. Bar scale: (**A**) 20 µm; (**B**) 50 µm; (**C**) 100 µm; (**D**) 10 µm.

**Figure 8 insects-13-00236-f008:**
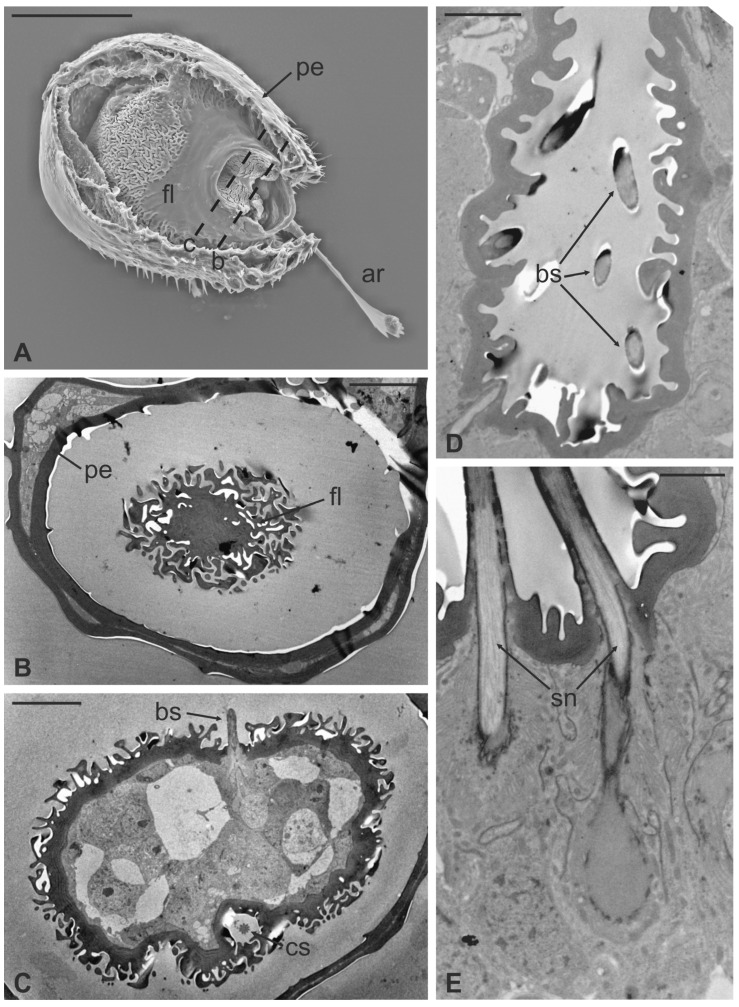
*Lipoptena fortisetosa*. (**A**) SEM ventral view of the antenna isolated from the head and partially opened. The pedicel (pe) surrounds the flagellum (fl). The anterior part of the flagellum bears the arista (ar) that protrudes externally from the pedicel; (**B**,**C**) TEM cross-section of the whole antenna taken according to the two dotted lines depicted in (**A**). (**B**) Antennal section carried out at the distal part of the pedicel (b line in **A**): the external structure, delineated by two cuticular layers is the pedicel, housing the flagellum with its elaborate and trabeculate cuticle; (**C**) flagellum, sectioned subapically (c line in **A**), showing some invaginations occupied by sensilla, as well as basiconic sensilla (bs), along with the typical cuticle; (**D**) detail of one of the cuticular cavities (sensory pit) occurring dorsally on the flagellum, accommodating several basiconic sensilla; (**E**) close-up view of the innermost part of a sensory pit: two basiconic sensilla are clearly visible with their innervating sensory neurons (sn). Bar scale: (**A**) 50 µm; (**B**,**C**) 10 µm; (**D**) 5 µm; (**E**) 2 µm.

**Figure 9 insects-13-00236-f009:**
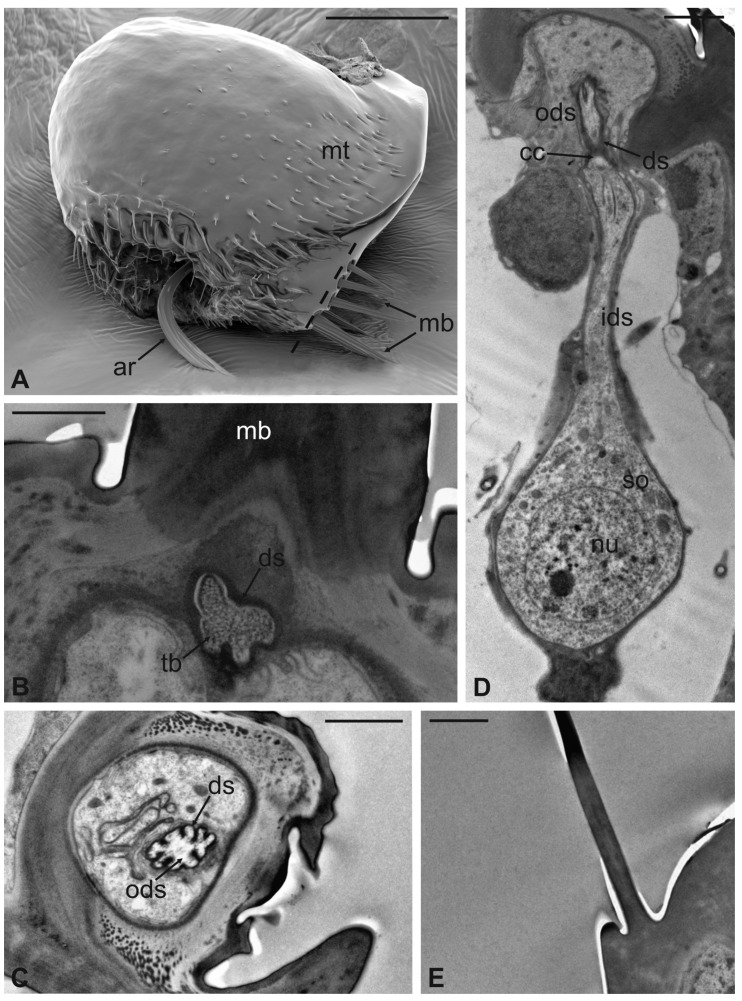
*Lipoptena fortisetosa*. (**A**) SEM dorsolateral view of the antenna: the arista (ar), the long and sharply pointed long bristles (mb), and numerous thin microtrichia (mt) are clearly visible; (**B**,**C**) TEM cross-sections of the whole antenna taken along the dotted line depicted in (**A**). The apical part of a single sensory neuron (tubular body, tb) that innervates the associated mechanosensory bristle is depicted. More proximally, the same neuron is visible at the level of the outer dendritic segment (ods), and it is encased by a thick dendrite sheath (ds); (**D**) TEM reconstruction obtained combining four different pictures showing the sensory neuron of the mechanosensory bristle. The cell body lies deeper in the antennal lumen. From the somata (so), a relatively short inner dendritic segment (ids) originates and, close to the cuticular wall, evolves in the outer dendritic segment (ods). The ciliary constriction (cc) region appears as a throttling, from which the dendrite sheath starts to be visible; (**E**) longitudinal section of a microtrichium showing no evidence of associated sensory neurons. Bar scale: (**A**) 50 µm; (**B**), 1 µm; (**C**–**E**) 2 µm.

**Figure 10 insects-13-00236-f010:**
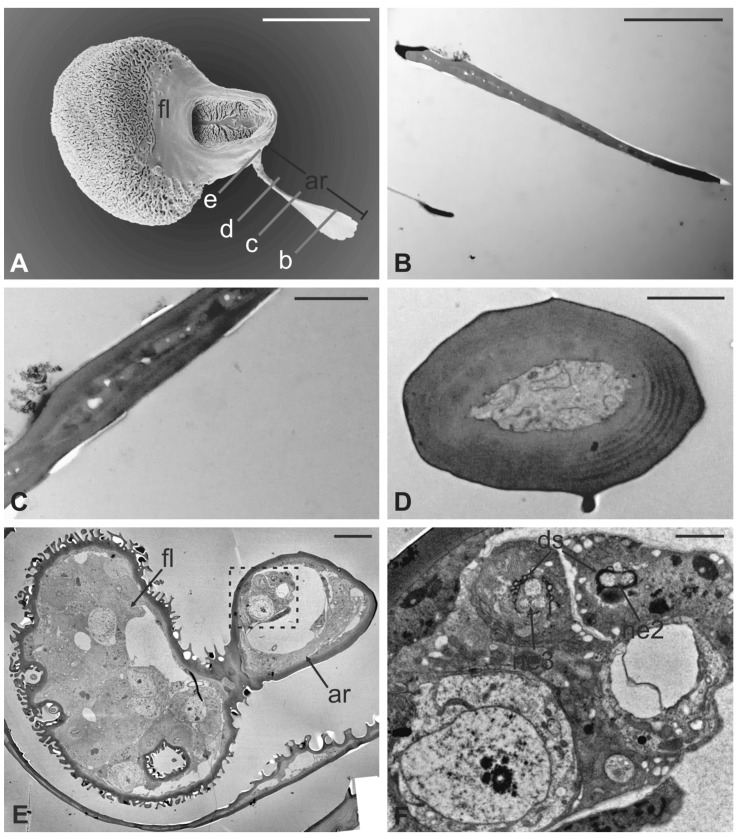
*Lipoptena fortisetosa* arista. (**A**) SEM ventral view of the flagellum (fl) removed from the remaining of the antenna showing the arista (ar). This is a long and slightly bended structure, with a flattened tip fan-shaped, supported by a thin, elliptical stalk; (**B**–**D**) TEM pictures showing cross-sections of the arista at the section plane levels as reported in (**A**). The arista appears flattened and without a perceptible internal lumen in (**B**,**C**), while in (**D**) the cuticle is thicker, and the small lumen is occupied by extracellular material; (**E**) TEM reconstruction obtained combining nine different pictures showing a cross-section of the region as reported in (**A**). At this level, the base of the arista is connected with the flagellum. The lumen of the arista displays a region (dotted square in (**E**) where (**F**) groups of neurons are visible. Two groups of neurons can be differentiated, one with three (ne3) and another with two neurons (ne2), in both cases enclosed by a dendrite sheath (ds). Bar scale: (**A**) 50 µm; (**B**) 10 µm; (**C**,**D**,**F**) 2 µm; (**E**) 10 µm.

**Figure 11 insects-13-00236-f011:**
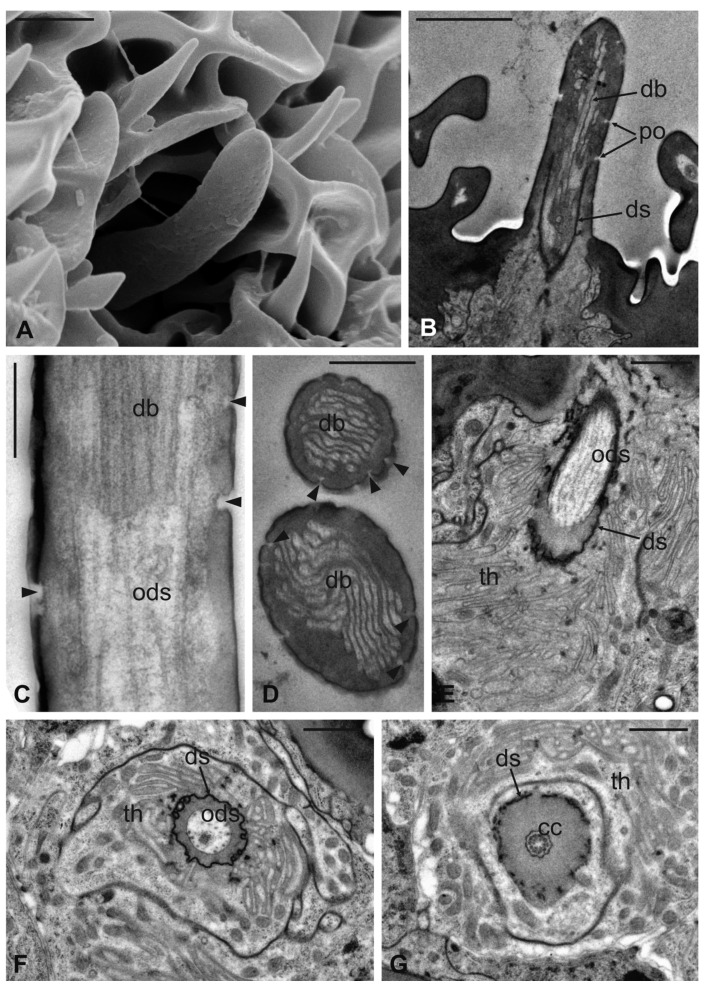
*Lipoptena fortisetosa* basiconic sensilla. (**A**) SEM picture showing one basiconic sensillum (bs) with its blunt tip and porous cuticle. The bs is surrounded by the elaborated cuticular sculpture of the flagellar wall. (**B**–**G**) TEM images of the bs; (**B**) longitudinal section of a bs characterized by a thin wall with many pores (po) and dendritic branches (db) entering the sensillum lumen. The dendrite sheath (ds) enters the sensillum and embeds the sensory neuron until it starts branching. The base of the sensillum is inflexibly inserted into the antennal wall; (**C**) detail of a longitudinal section of a bs displaying the unbranched outer dendritic segment (ods) of the sensory neuron that starts forming dendritic branches (db). Some pores are visible (arrowheads); (**D**) cross-section showing two bs sectioned at their distal region: clusters of dendritic branches filling the lumen are visible, as well as several cuticular pores. (**E**) Longitudinal section of a bs taken at the base level unveiling a single sensory neuron embedded by the dendrite sheath and numerous microvilli belonging to the thecogen cell (th). (**F**–**G**) Cross-sections of the bs sensory neuron; (**F**) the outer dendritic segment surrounded by the dendrite sheath and the thecogen cell; (**G**) the same sensory neuron depicted more proximally, at the ciliary constrictions (cc) level, where the dendrite sheath originates. Bar scale: (**A**,**B**) 2 µm; (**C**,**D**) 0.5 µm; (**E**–**G**) 1 µm.

**Figure 12 insects-13-00236-f012:**
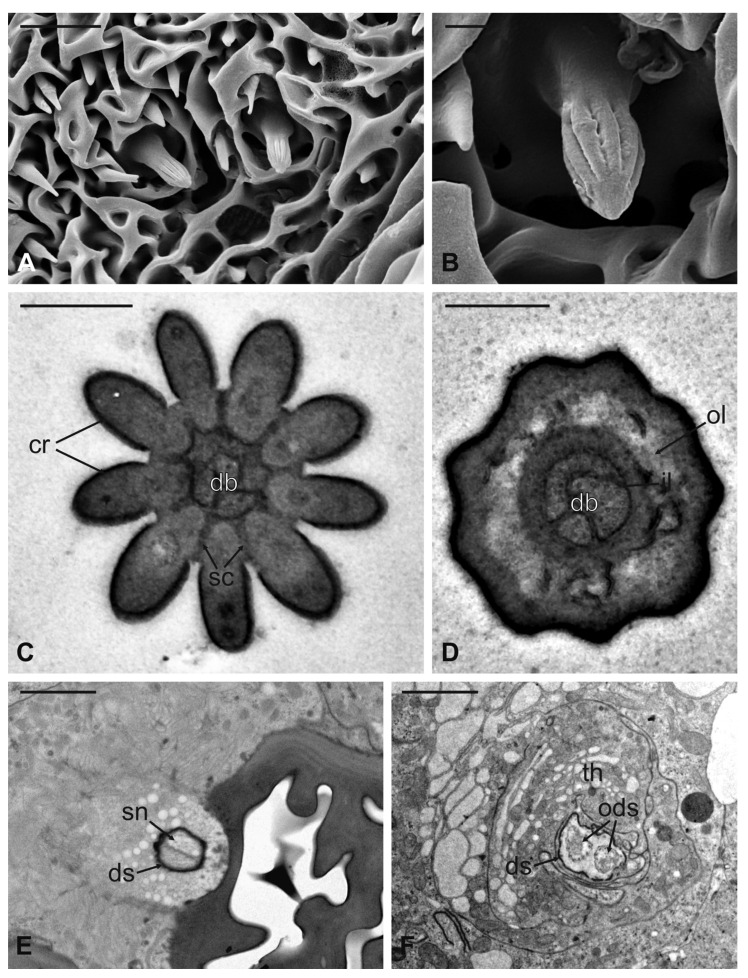
*Lipoptena fortisetosa* coeloconic sensilla. (**A**–**B**) SEM pictures showing two coeloconic sensilla (cs) inserted on the flagellar wall. These sensilla exhibit distinct longitudinal ridges that define correspondent longitudinal grooves (**B**) originating at the distal half of the peg and reaching the tip where they converge and merge together; (**C**,**D**) serial TEM sections of a single cs; (**C**) distal section of the cs, at the level of the grooves. The typical stellate structure related to the presence of cuticular ridges (cr) is visible. Between each of the cuticular ridges, a spoke channel (sc), connecting the outside with the sensillum lumen, is visible. At this level, the sensillum lumen shows tree dendritic branches (db) of sensory neurons; (**D**) proximal section of the same sensillum where the cuticular ridges are not present. The internal double-walled organization of the cuticle appears clear at this level, defining an outermost (ol) and an innermost lumen. This latter shows three dendritic branches; (**E**) cross-section at the base of the cs, where a thick dendrite sheath (ds) embeds two sensory neurons (sn); (**F**) cross-section of the two sensory neurons belonging to cs. At this level the dendrite sheath appears less compact and not so tightly associated with the outer dendritic segments (ods). The thecogen cell (th) envelops both structures. Bar scale: (**A**) 5 µm; (**B**) 1 µm; (**C**,**D**) 0.5 µm; (**E**,**F**) 2 µm.

## Data Availability

The data that support the findings of this study consist of pictures obtained through SEM and TEM observations. Selected pictures are displayed in the article. All microscopy data are available from the authors upon request.

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
