# Peer review of "Antennal Morphology and Fine Structure of Flagellar Sensilla in Hippoboscid Flies with Special Reference to Lipoptena fortisetosa (Diptera: Hippoboscidae)"

_insects, 2022, doi:10.3390/insects13030236_

Round 1

Reviewer 1 Report

The morphology and sensory structure of antennal appendages of four hippopotamuses were studied. Typical conformations of antennae surrounded by the third segment (flagella) were observed in two of them. In addition, two types of receptors have been detected and their role in sensing host odor and carbon dioxide has been assumed. In addition, other antennal structures seem to be involved in the detection of temperature and humidity changes. The purpose of this study is to study the antennae of four species of hippopotamuses, namely, lipoptera cervi (Linnaeus, 1758), lipoptera fortisetosa MAA (1965), hippobosca equina Linnaeus (1758) and pseudolynchia canariensis (Macquart, 1840), Study the morphology and sensory structure of these appendages. The typical conformation of the antennae surrounded by the third segment (flagella) in the first two was observed. In addition, two types of receptors have been detected and their role in sensing host odor and carbon dioxide has been assumed. In addition, other antennal structures seem to be involved in the detection of temperature and humidity changes. Our findings confirm that these hippos use chemoreceptors at the host location, thus providing an in-depth understanding of this complex process, but there is little research in this population. This work is helpful to help us understand the host localization of extramural blood sucking insects of hippocampal family. The manuscript is well written, scientific and reasonable. Before being accepted for publication, in order to improve the quality of this work, some minor modifications need to be made below.

Special comments:

Line 120: “All hippoboscid adults have been anaesthetized for 20 min at −20°C for 20 min and 120 .” please specify the number of samples.

Line 226: You said that there are strong mechanical bristles arranged on the edge of the pedicel. What is the measure of strong mechanical feeling, structure or stiffness? Please elaborate further.

Line 472-473: “…H. equina displays a higher number of sensory pits compared to the other three species…”,please supplement the data description.

Line 490: “In D. melanogaster coeloconic grooved sensilla are involved in the detection of ammonia, ketones, and amines;” please mark references or specific descriptions.

Line539-542: You said that the significant decrease in the number and size of basal receptors and the number of related sensory neurons may be related to the reduction of the utilization range of volatiles in intraspecific and interspecific interactions. Is it related to the sensitivity of the basal sensor?

Reviewer 2 Report

The manuscript describes the antennal morphology of 4 species of hematophagous ectoparasites hippoboscid flies using SEM and TEM. The paper is well written with nice layout of the results as well as a thorough discussion of the finds. The TEM study of the sensilla in Lipoptena fortisetosa was well done. My main concern with the paper is the low resolution images of the panels which were due to the lossy PDF conversion which is really frustrating to look for reviewing. I suggest the authors to understand a bit more about the PDF conversion and convert the image appropriately. I am certain it would be corrected in the production phase of publications. Additionally, there were some discussions on the numbers, distributions of the sensilla among the investigated 4 species which correlated to the their evolution/lifestyles (~Line 470-480), a table of the numbers/distributions of the 2 types of sensilla would be nice since it is not very difficult to obtain with such study instead of referring as “abundant” or “fewer” which are hardly scientific (page 17 first paragraph).  

Here are some minor details:

Line 24-26: “Our findings allowed to confirm that these hippoboscids use the chemoreception in the host location, giving insights on this complex process (which is) poorly investigated in this group.”

Line37-42:“These structures could aid convey volatile compounds towards the flagellar sensory area. However, peculiar sensory neurons characterize the unarticulated arista which could be able to detect temperature variations. Coeloconic sensilla could be involved in a thermo- hygro- and carbon dioxide reception at long-distance, while the poorly porous basiconic sensilla could play a role in the host odor perception at medium-short distance.”  While discussed, this study only observes the ultrastructure of the sensilla and there is no functional/nor behavioral data supporting any of claims. I would recommend to reword the sentences into something of less assertive.  

Figures: Beside the loss of resolution from PDF compression mentioned above, generally speaking, the figures were annotated with letters only. It would be helpful for no-expert readers with arrows indicating important points (e.g. Figure 4, arrows for panels C and D would be helpful).

Figure 10: The panel F is shown as mirror image or a rotation of panel E? The resolution loss in the figure is so severe that I could not really tell for sure. But I think it was flipped. It would be nice to show the magnified area at the same orientation as the dashed box in panel E.

Line 125- In Materials and Methods It seems that the antennae were “excised from the heads after 99% ethanol dehydration and dissected to extract the internal flagella.” This is a bit perplexing as after 99% dehydration, the tissue usually becomes much more brittle and easily damaged. It would be much easier to perform such procedure in earlier stages. Please verify and clarify it was done as described.

Reviewer 3 Report

This is a nice and straightforward manuscript that delves into very detailed morphology of antennal structures of hippoboscid flies. I recommend publication after a number of corrections to the language. They have found a unique antennal feature, and this sort of basic morphological investigation lays the foundation for additional research questions and experiments. Basic work like this should be encouraged.

There is a need for extensive editing for correct English. See the comments in the attached Word file.

Round 2

Reviewer 1 Report

I read the paper.All my concerns have been well organized. For my part, I accept the paper.